# The real-world outcomes of multiple myeloma patients treated with daratumumab

Agoston Gyula Szabo[1]☯, Tobias Wirenfeldt Klausen[2]☯, Mette Bøegh Levring[3], Birgitte Preiss[4], Carsten Helleberg[2], Marie Fredslund Breinholt[5], Emil Hermansen[6], Lise Mette Rahbek Gjerdrum[7], Søren Thorgaard Bønløkke[8], Katrine Nielsen[8], Eigil Kjeldsen[8], Katrine Fladeland Iversen[1], Elena Manuela Teodorescu[9], Marveh Dokhi[10], Eva Kurt[11], Casper Strandholdt[12], Mette Klarskov Andersen[13], Annette Juul Vangsted[10]*

1 Department of Hematology, Vejle Hospital, Vejle, Denmark, 2 Department of Hematology, Herlev University Hospital, Herlev, Denmark, 3 Department of Hematology, Odense University Hospital, Odense, Denmark, 4 Department of Pathology, Odense University Hospital, Odense, Denmark, 5 Department of Pathology, Herlev University Hospital, Herlev, Denmark, 6 Department of Hematology, Zealand University Hospital, Roskilde, Denmark, 7 Department of Pathology, Zealand University Hospital, Roskilde, Denmark, 8 Department of Hematology, Aarhus University Hospital, Aarhus, Denmark, 9 Department of Hematology, Aalborg University Hospital, Aalborg, Denmark, 10 Department of Hematology, Rigshospitalet, Copenhagen University, Copenhagen, Denmark, 11 Department of Hematology, Regionshospitalet Holstebro, Holstebro, Denmark, 12 Department of Hematology, Esbjerg Hospital, Esbjerg, Denmark, 13 Department of Clinical Genetics, Rigshospitalet, Copenhagen, Denmark

☯ These authors contributed equally to this work.
* annette.juul.vangsted@regionh.dk

**Data Availability Statement:** All relevant data are within the manuscript and its Supporting information files.

## Abstract

Most patients cannot be included in randomized clinical trials. We report real-world outcomes of all Danish patients with multiple myeloma (MM) treated with daratumumab-based regimens until 1 January 2019.

### Methods

Information of 635 patients treated with daratumumab was collected retrospectively and included lines of therapy (LOT), hematologic responses according to the International Myeloma Working Group recommendations, time to next treatment (TNT) and the cause of discontinuation of treatment. Baseline characteristics were acquired from the validated Danish Multiple Myeloma Registry (DMMR).

### Results

Daratumumab was administrated as monotherapy (Da-mono) in 27.7%, in combination with immunomodulatory drugs (Da-IMiD) in 57.3%, in combination with proteasome inhibitors (Da-PI) in 11.2% and in other combinations (Da-other) in 3.8% of patients. The median number of lines of therapy given before daratumumab was 5 for Da-mono, 3 for Da-IMiD, 4 for Da-PI, and 2 for Da-other. In Da-mono, overall response rate (ORR) was 44.9% and median time to next treatment (mTNT) was 4.9 months. In Da-IMiD, ORR was 80.5%, and mTNT was 16.1 months. In Da-PI, OOR was 60.6% and mTNT was 5.3 months. In patients treated

**Funding:** We have received funding from The Danish Cancer Society, an independent democratic membership organization, whose course is charted by the volunteers and members. From The Danish Cancer society my department received financial support for me being off clinical work one day per week to do the scientific work presented here. Furthermore, we paid for statistical analysis and for collection of clinical data from all haematological department in Denmark. The funders had no role in study design, data collection and analysis, decision to publish, or preparation of the manuscript.

**Competing interests:** The authors have declared that no competing interests exist.

with Da-other, OOR was 54,2% and mTNT was 5.6 months. The use of daratumumab in early LOT was associated with longer TNT (p<0.0001). Patients with amplification 1q had outcome comparable to standard risk patients, while patients with t(4;14), t(14;16) or del17p had worse outcome (p = 0.0001). Multivariate analysis indicated that timing of treatment (timing of daratumumab in the sequence of all LOT that the patients received throughout the course of their disease) was the most important factor for outcome (p<0.0001).

## Conclusion

The real-world outcomes of multiple myeloma patients treated with daratumumab are worse than the results of clinical trials. Outcomes achieved with daratumumab were best when daratumumab was used in combination with IMIDs and in early LOT. Patients with high-risk CA had worse outcomes, but patients with amp1q had similar outcomes to standard-risk patients.

## Introduction

The excellent outcomes of patients treated with daratumumab in randomized clinical trials (RCTs) have changed the treatment strategy of multiple myeloma (MM). However, RCTs are conducted in selected patient populations according to strict inclusion and exclusion criteria. Several studies have shown that a significant part of patients with MM are ineligible for clinical trials and that these patients have worse OS [1–5]. The clinical efficacy of daratumumab-based therapy in most MM patients is therefore not well described, and real-world data are warranted. Daratumumab monotherapy (Da-mono) was approved in 2016 for patients who had received at least two previous lines of therapy. Daratumumab combination regimens were approved in 2017 for the treatment of MM patients at first relapse.

In this study, we report the clinical outcomes of patients treated with daratumumab-based therapy in a complete nationwide cohort. We performed detailed assessment of the entire clinical course of every patient in Denmark, who had initiated treatment with daratumumab-based regimens until 1 January 2019. The aims of our study were to report response rates and TNT in patients treated with daratumumab monotherapy (Da-mono), daratumumab in combination with immunomodulatory agents (Da-IMiD), and daratumumab in combination with proteasome inhibitors (Da-PI). Secondly, we wanted to assess the influence of timing of the first daratumumab-containing line of therapy on patient outcomes. Thirdly, we wanted to report causes of discontinuation of daratumumab treatment. Lastly, we wanted to focus on the outcomes of patients with the high-risk cytogenetic abnormalities (CA) t(4;14); t(14,16), del (17p) and amp1q. The study was approved by the Danish Data Protection Agency (18/22825) and the Danish Patient Safety Authority (3-3013-2047/2r). The ethics committee waived the requirement for informed consent.

## Materials and methods

The study was approved by the Danish Data Protection Agency (18/22825) and the Danish Patient Safety Authority (3-3013-2047/2r). The ethics committee waived the requirement for informed consent.

In Denmark, the access to healthcare services is universal, population-based and publicly financed. Available treatment strategies are presented in S1 Table in S1 File. We can therefore

present outcome of daratumumab monotherapy in many patients. Lenalidomide maintenance therapy after ASCT was approved after data cut-off for this study.

Patients treated with daratumumab were identified from the pharmacy registries of the participating departments. The study cut-off was 1 January 2019. Individual treatment data were collected by retrospective review of patient records performed by trained physicians from hematology departments in Denmark (Aalborg, Aarhus, Esbjerg, Herlev, Holstebro, Odense, Rigshospitalet, Roskilde, and Vejle). Data represent a population-based cohort of all patients treated with daratumumab in Denmark between 1 June 2018 and 1 November 2019. Data were entered in a Research Electronic Data Capture (REDcap) database (S2 Table in S1 File). Queries were sent for data completeness and corrections. Baseline characteristics were acquired from the validated Danish Multiple Myeloma Registry (DMMR), which includes clinical data for every patient diagnosed with MM since 2005 [6]. The dataset from the DMMR was merged with the study database Lines of therapy (LOT) and hematologic responses were evaluated according to the International Myeloma Working Group (IMWG) recommendations [7, 8]. For each LOT, the date of initiation, the used regimen, the hematologic responses, the date of discontinuation and the cause of discontinuation were registered. Toxicities were registered only if they were the cause of discontinuation of a LOT. Bone marrow biopsies for response assessment are not routinely performed in Denmark outside of clinical trials except for patients treated with high-dose melphalan with autologous stem cell transplantation (HDM-ASCT), therefore, responses were aggregated into three response categories for data analysis: very good partial response or better (≥VGPR), partial response (PR) and minimal response or worse (≤MR). Overall response rate was calculated by combining rates of ≥VGPR and PR. In case of discontinuation due to toxicity, the nature of the toxicity was specified in further detail. TNT was used as outcome parameter as the dates of initiation of a new LOT were uniformly registered in all patients. TNT was defined as the time from the date of initiation of a LOT until either the date of initiation of the subsequent LOT, the date of death or the date of last follow-up in patients still on the last LOT (S1 Fig in S1 File). Timing of treatment was the timing of daratumumab in the sequence of all LOT that the patients received throughout the course of their disease. Fluorescence in situ hybridization (FISH) data of CA were reviewed and registered by experienced consultants in cytogenetic analysis. When consecutive FISH analysis was done, the results from the most recent assessment before first exposure to daratumumab was used. The cut-off for chromosome deletions, chromosome translocations and chromosome amplifications were their presence in at least 10% of tumor cells. The CA t (4;14), t(14;16), and del(17p) were defined as high-risk CA. Based on the first daratumumab-containing LOT (Da), the patient cohort was arranged into three subgroups: patients treated with Da-mono, Da-IMiD or Da-PI.

## Statistical analysis

Categorical variables where presented with number and percentages and compared between groups by Chi-square tests or Fisher's exact test in the case of small numbers. Continuous variables were presented with median and interquartile range (IQR). Continuous variables were compared between multiple groups by Kruskal-Wallis tests and between two groups by Mann-Whitney test. Time to next treatment were presented by Kaplan-Meier curves. Median times and proportions at specific times were extracted from the Kaplan-Meier statistics and presented with 95% confidence interval (CI). Differences between groups were calculated by log-rank tests. Furthermore, a Cox proportional hazard model was calculated and hazard ratios (HR) with 95% confidence intervals were presented. To find risk factors correlated to TNT, a univariate and a multivariable Cox proportional hazard model were applied. Only significant

variables from the univariate models were entered in the multiple model. Time to follow up was calculated with the reverse Kaplan-Meier method. All p-values were two-sided and p-values $\leq 0.05$ were considered statistically significant. R version 3.6.1 were used for all calculations.

# Results and discussion

## Patient characteristics and subgroups

Six hundred and thirty-five patients were treated with daratumumab. Of these, 225 patients (35.4%) were still on treatment with Da at data cut-off. Patients were diagnosed with MM between 1986 and 2018. The median age for all patients at diagnosis was 66 (interquartile range (IQR): 58–71) years, and the median age at initiation of Da was 70 years (IQR: 63–75). The median follow-up from start of Da was 18.0 months. The cohort consisted of 357 men and 278 women. The CA t(4;14), t(14;16), del(17p), and amp1q were assessed in 73.5%, 72,1%, 76.4% and 73.7% of patients, and were positive in 13.9%, 4.1%, 13.6% and 29.5% of patients, respectively (S3 Table in S1 File). A high-risk CA was present in 23.5% of patients still on Da as compared to 32.2% of patients who started a new LOT after Da (p = 0.048). Overall, patients received a median of 5 (range: 1–22) lines of therapy throughout the course of their disease (S4 Table in S1 File). Ten patients received daratumumab in combination with lenalidomide in first line as part of the Maia study. The outcome of these patients is presented in Fig 2. Three hundred and nine patients (48.7%) were treated with up-front HDM-ASCT within the first year from diagnosis. Of these, 254 (82%) received HDM-ASCT within 6 months from diagnosis.

Baseline characteristics of patients treated with Da-mono (176 patients), Da-IMiD (364 patients), Da-PI (71 patients), or Da-other (24 patients) are shown in summary in Table 1 and in S5 Table in S1 File. We compared the treatment groups Da-mono, Da-IMiD and Da-PI. Patients treated with Da-mono were older, fewer were treated with up-front HDM-ASCT, and fewer had elevated lactate dehydrogenase (LDH) at diagnosis. The median number of prior LOT was 5 in Da-mono, 3 in Da-IMiD, and 4 in Da-PI (p<0.0001). The percentage of patients exposed to all four drugs (quadruple-exposed) prior to Da was 18.2% in Da-mono, 3,7% in Da-IMiD and 7.0% in Da-PI. In most of the quadruple-exposed patients, Da was given as a $\geq 6^{th}$ LOT (S8 Table in S1 File).

## The use of daratumumab

A detailed description of the use of daratumumab is presented in S5, S8 Tables and S2 Fig in S1 File. The most frequently used regimen was Da-IMiD, except in sixth or later lines, where Da-mono was most commonly used. Da-PI was primarily used as a 3. line therapy and in 6. or later lines.

## Response to daratumumab

The ORR to Da was 67.2%, including 42% $\geq$VGPR and 25.2% PR. Response rates with the different daratumumab-based regimens are shown in S7 Table (S1 File). The ORR and $\geq$VGPR rates were 44.9% and 19.3% in Da-mono, 80.5% and 56% in Da-IMiD, and 60.6% and 32.4% in Da-PI, respectively (p<0.0001). S3 Fig in S1 File illustrates $\geq$VGPR, PR and $\leq$MR for all treatment combinations. Increase in $\geq$VGPR rates was observed when Da was administered as early as possible (S7 Table and S4 Fig in S1 File). In patients with high-risk CA, the ORR and $\geq$VGPR rates were 32.3% and 12.9% in Da-mono, 72.8% and 51.9% in Da-IMiD, and

**Table 1. Summary of characteristics and outcome of 635 patients treated with daratumumab.**

| N = 635 | Da-mono | Da-IMiD | Da-PI | Da-other | p value[1] |
|---|---|---|---|---|---|
| N (%) | 176 (27.7) | 364 (57.3) | 71 (11.2) | 24 (3.8) | |
| Age at start of treatment; median IQR | 72 (67–77) | 70 (63–74) | 70 (62–75) | 63 (51–74) | 0.002 |
| Gender male/female; (% male) | 105/71 (59.7) | 201/163 (55.2) | 37/34 (52.1) | 14/10 (41.7) | 0.48 |
| ISS | | | | | 0.64 |
| III; no (%) | 42 (29.8) | 99 (32.0) | 18 (29.5) | 7 (35.0) | |
| High-risk CA: at least one of del17p, t(14;16), t(4;14);N (%) | 31 (24.8) | 81 (30.1) | 18 (36.0) | 5 (26.9) | 0.30 |
| [N missing] | [51] | [95] | [21] | | |
| Amp1q; no (%) | 40 (32.0) | 79 (28.9) | 14 (27.5) | 5(26.9) | 0.77 |
| [N missing] | [51] | [91] | [20] | | |
| Prior treatment before daratumumab | | | | | |
| HDM-ASCT in first line | 59 (33.5) | 190 (52.2) | 41 (57.7) | 19 (79.2) | <0.001 |
| Bortezomib; N (%) | 166 (94.3) | 340**(96.0) | 66 (93.0) | 24 (100) | 0.44 |
| Lenalidomide; N (%) | 157 (89.2) | 158**(44.6) | 56 (78.9) | 18 (75.0) | <0.0001 |
| Carfilzomib; N (%) | 57 (32.4) | 72**(20.3) | 14 (19.7) | 6 (25.0) | 0.006 |
| Pomalidomide; N (%) | 78 (44.3) | 35**(9.9) | 16 (22.5) | 7 (29.2) | <0.0001 |
| IMiDs and PI; N (%) | 158 (89.8) | 159**(44.9) | 55 (77.5) | 19 (79.2) | <0.0001 |
| Quadruple-exposed; N (%) | 32 (18.2) | 13**(3.7) | 5 (7.0) | 3 (12.5) | <0.0001 |
| Number of lines of therapy given before daratumumab; median (range) | 5 (2–16) | 3 (1–11) | 4 (2–10) | 2 (1–3) | <0.0001 |
| Time from diagnosis to start of daratumumab; mo; median (IQR) | 48.2 (26.8–79.8) | 37.9 (18.1–69.4) | 49.2 (30.2–91.1) | 28.5 (7.7–86.7) | 0.002 |
| Median follow-up after start of daratumumab; mo | 22.7 | 17.8 | 20.3 | 27.7 | |
| ORR, N (%) | 79 (44.9) | 293 (80.5) | 43 (60.6) | 13 (54.2) | <0.0001 |
| ≥VGPR, N (%) | 34 (19.3) | 204 (56.0) | 23 (32.4) | 6 (25.0) | <0.0001 |
| TNT all lines, median mo (CI)[N] | 4.9 (3.7–5.8) [176] | 16.1 (13.7–20.3) [364] | 5.3 (3.5–8.2) [71] | 5.6 (2.8–12.7) [24] | <0.0001 |
| OS, median and CI | 28.2 mdr (19.8–38.0) | 33.2 (25.8-NR) | 25.2 (17.8-NR) | 16.3 (9.0-NR) | <0.0001 |

N = number; IQR: interquartile range; mo: months. Quadruple-exposed = previously treated with both bortezomib, lenalidomide, pomalidomide and carfilzomib.

** without the 10 patients that received daratumumab at first line.

[1]The p-value describes the difference between Da-IMiDs, Da-PI, and Da-mono.

44.4% and 22.2% in Da-PI, respectively (p = 0.0002; S6 Table in S1 File). Among the 225 patients still on Da, 73.3% had ≥VGPR and 19.1% had PR at the time of cut-off.

## Time to next treatment according to regimens and timing

The median TNT of Da was 9.8 months (95% CI: 8.4–11.8). The TNT and OS in different daratumumab-based regimens is shown in Fig 1. The median TNT was significantly longer for Da-IMiD compared to Da-mono, Da-PI and Da-other (p<0.0001) and a trend to better OS was found p = 0.08. Earlier use of Da was associated with longer TNT (Fig 2 and S6 Table in S1 File). Daratumumab monotherapy was predominantly used as 4. or later LOT. The median TNT for Da-mono decreased from 9.2 months in 4. line to 3.9 months in 6. and later LOT. Da-IMiD was predominantly used as a 2. LOT. The median TNT for Da-IMiD decreased from 25.9 months in 2. line to 6.4 months in 6. and later LOT. Da-PI was mainly used in 3. or later LOT. The median TNT for Da-PI decreased from 5.5 months in 3. line to 3.7 months in 6. and later LOT.

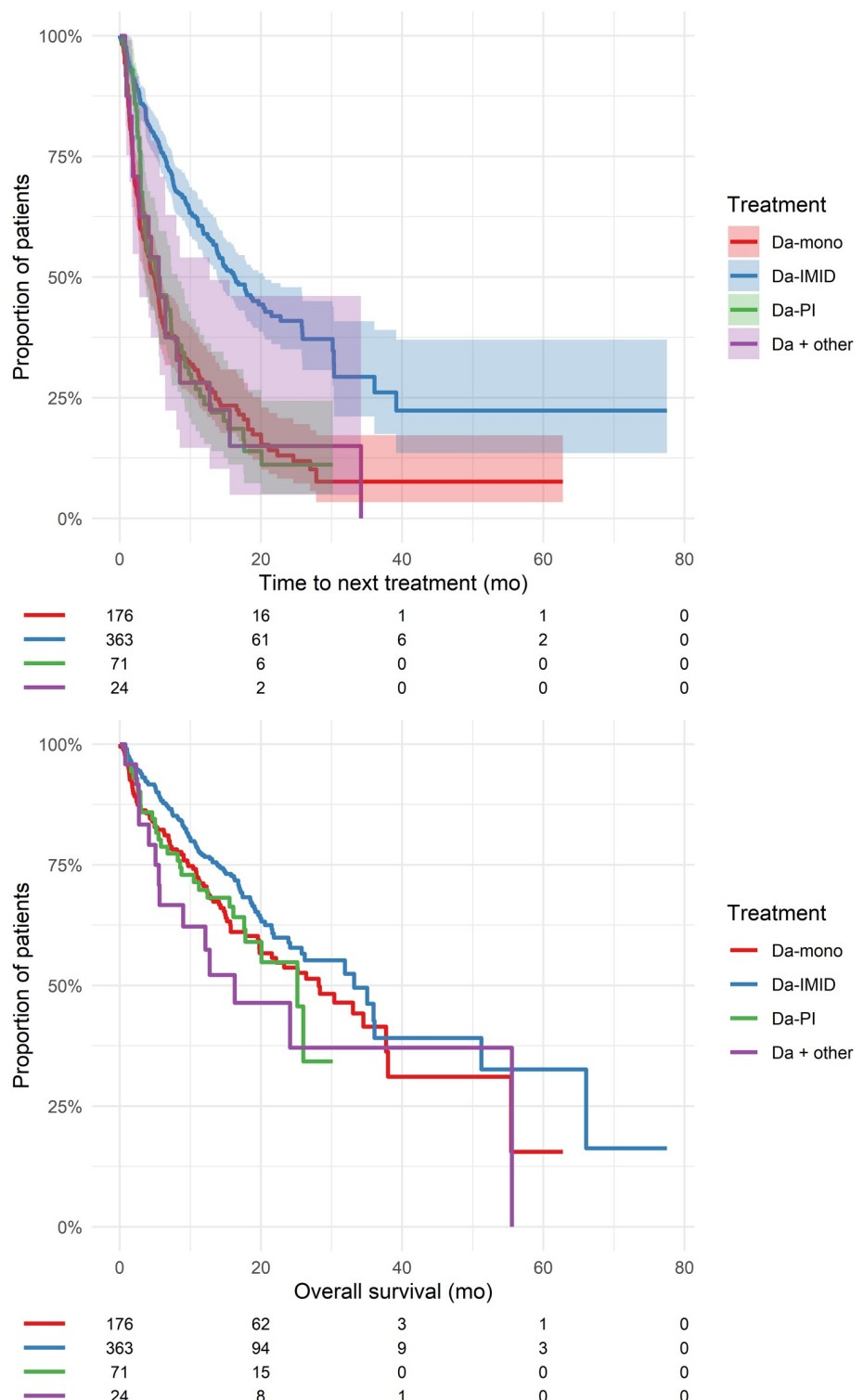

**Fig 1. Time to next treatment and overall survival of the first daratumumab therapy.** Patients at risk is shown below the figure. Abbreviations: Da-other = daratumumab in other combinations than IMiD and PI; mo = months.

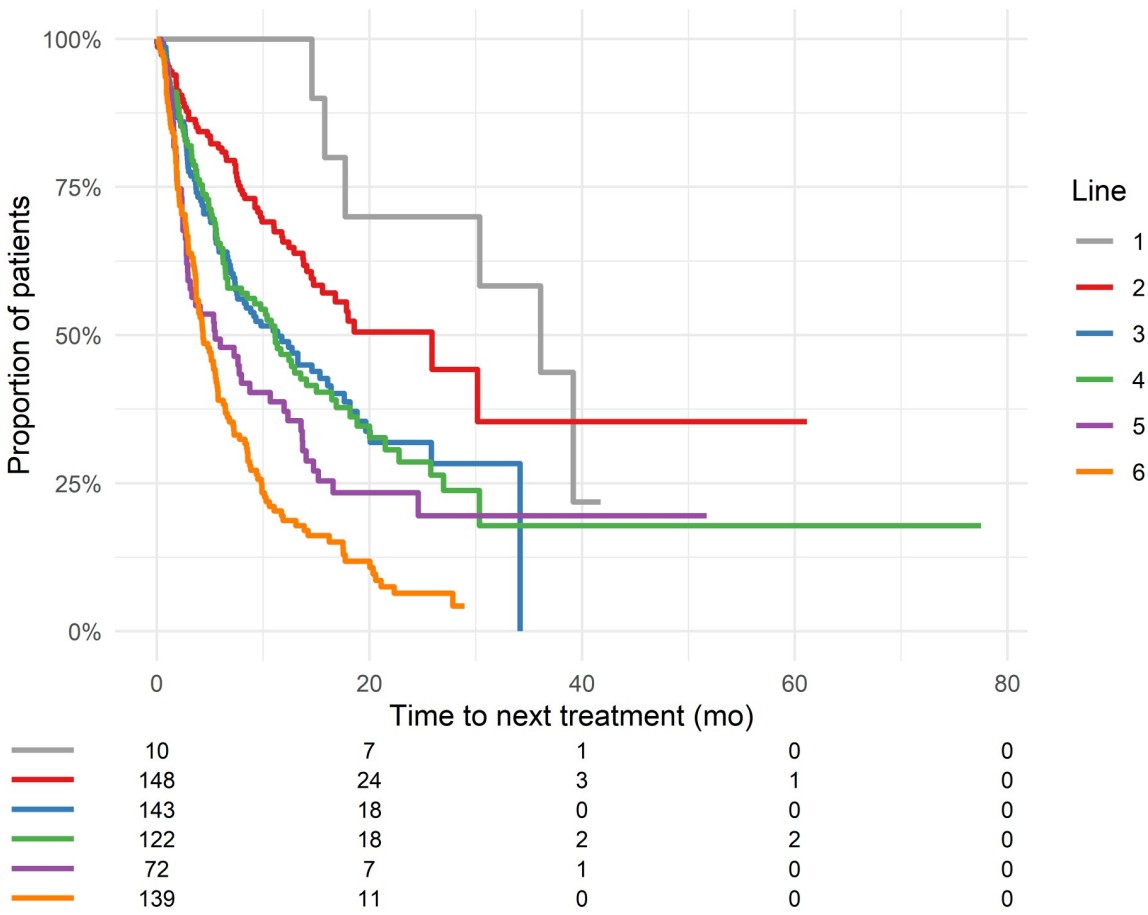

**Fig 2. Time to next treatment depending on the timing of daratumumab therapy.** Patients at risk is shown below each figure. A: TNT for all combinations of daratumumab depending on timing. Cox-regression analysis showed longer TNT in early lines (p<0.0001).

## Time to next treatment according to cytogenetic abnormalities

The TNT of daratumumab of patients with standard-risk, high-risk CA +/- amp1q are shown in Fig 3 and S9 Table in S1 File. The median TNT of Da was 7.6 months (CI: 5.6–11.9) in patients with a high-risk CA, 9.8 months (CI: 6.5–16.9) in patients with amp1q, and 11.7 months (CI: 9.5–15.6) in patients with standard-risk CA. The median TNT for patients with a high-risk CA was 3.7 months in Da-mono, 11.8 months in Da-IMiD and 4.0 months in Da-PI (p = 0.002; S5 Table in S1 File). Earlier timing of Da resulted in longer TNT in patients with standard-risk CA (p<0.0001), while the association was not significant in patients with high-risk CA (p = 0.07; S6A and S6B Fig and S10 Table in S1 File). However, the TNT for 2. line was 11.8 months compared to 4.2–6.6 months in 3. line and later LOT indicating that the lack of significance may be caused by the low number of patients with high-risk CA. The TNT of patients with amp1q was not different than the TNT of patients with standard-risk CA but TNT of patient with am1q was longer as compared to patients carrying high-risk CA combined and amp1q (P = 0.036; S10 Table in S1 File).

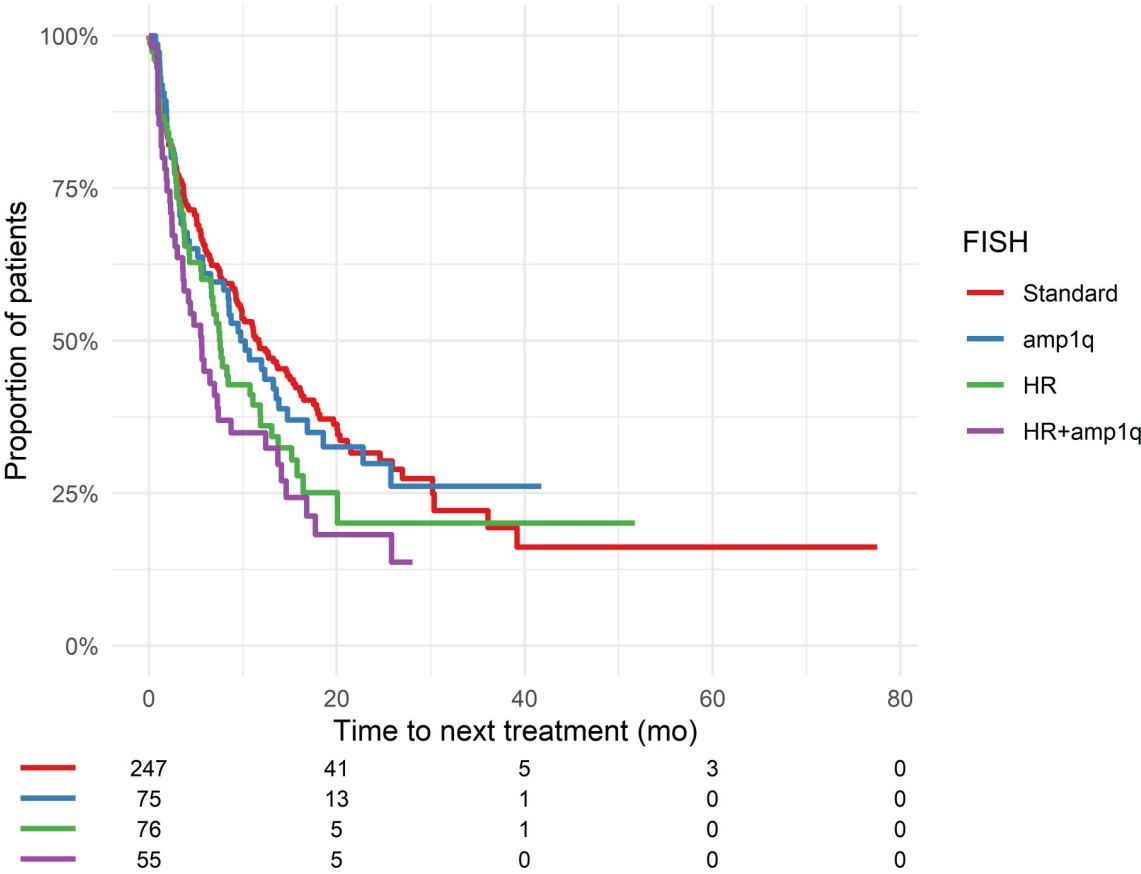

**Fig 3. Time to next treatment of the first daratumumab therapy depending on cytogenetic abnormalities.** Patients at risk is shown below each figure. High-risk CA was defined as the presence of del17p, t(4:14) or t(14:16). Abbreviations: mo = months. A:TNT for for all combinations of daratumumab-containing line of therapy depending on CA. Compared with patients with standard-risk CA, patients with amp1q had similar TNT (p = 0.65). Patients with high-risk CA had a trend towards shorter TNT (p = 0.003).

### Time to next treatment according to treatment with or without daratumumab

We explored TNT in daratumumab-based regimens with the TNT of regimens that did not include daratumumab in the same LOT. Daratumumab-based regimens resulted in longer TNT from 2. to 4. LOT, while the addition of daratumumab in the 5. and later LOT had no significant effect on TNT (S12 Table and S7 Fig in S1 File). In this non-randomized setting, the median TNT for Da compared to the same LOT without daratumumab was 25.9 versus 11.1 months in the 2. LOT 11.4 versus 8.8 months in the 3. LOT and 11.1 versus 7.7 months in the 4. LOT.

### Univariate and multivariate analysis of factors affecting time to next treatment

We explored univariate and multivariate analysis of factors affecting TNT of Da, as presented in S13 Table in S1 File. Univariate analysis included timing of Da (p<0.0001), the daratumumab-based regimen (p<0.0001), HDM-ASCT in first LOT (p = 0.92), age (p = 0.34), IgA M-protein isotype (p = 0.037), high-risk CA (p = 0.005) and amp1q (p = 0.19). In multivariate

analysis, timing of Da (p<0.0001) and the daratumumab-based regimen (p<0.0001) had a significant effect on TNT. A borderline significance was found for high-risk CA (p = 0.052).

## Reasons for discontinuation of daratumumab

Four hundred and ten patients discontinued Da (S14 Table in S1 File). The three most frequent causes of discontinuation of Da were progressive disease in 245 patients (59.8%), toxicity in 56 patients (13.7%) and insufficient response in 49 patients (12.0%). Progressive disease was the most frequent cause of discontinuation in all treatment groups. In patient´s treated with Da-mono, Da-IMiD, and Da-PI, 3.4%, 9.6%, and 15.5% discontinued treatment due to toxicity, respectively. The most frequent toxicities causing discontinuation of Da were infections (30.4%), peripheral neuropathy (16.1%), bone marrow suppression (10.7%), and gastrointestinal symptoms (10.7%), as shown in S15 Table in S1 File.

## Discussion

Our work presents data from all Danish myeloma patients treated with daratumumab-based therapy.

The outcomes of patients treated with Da-mono were slightly better than those reported in the GEN501 and SIRIUS studies where daratumumab was used as monotherapy [9–11]. The ORR to Da-mono in our study was higher (44.9% versus 31.1%) while the TNT was comparable to the previously reported results (median TNT 4.9 versus median PFS 4.0 months) [11]. Compared with the GEN501 and SIRIUS studies, patients in our cohort were older (median 72 versus 64 years), the percentage of patients with ISS III stage disease was lower (29.8% versus 38%), fewer patients had been treated with HDM-ASCT (33.5% versus 78%), fewer patients had received >3 LOT prior to daratumumab treatment (54.0% versus 76%) and less patients were quadruple-exposed (18.3% versus 31%). In summary, we believe that timing of treatment may explain the slightly better real-world outcomes in our cohort of patients than in in the GEN501 and SIRIUS studies [9–11].

In the POLLUX study, where daratumumab was combined with lenalidomide and dexamethasone, the TNT was considerably longer compared to our cohort of patients (56.6 versus 16.1 months) [12, 13]. ORR was 92.9% in the POLLUX study compared to 80.5% in our cohort [12, 13]. Patients in our cohort were older (median 70 versus 65 years), had higher percentage of ISS stage III disease (32% versus 19.6%), more patients had high-risk CA (30.1% versus 15.4%), fewer patients were previously treated with HDM-ASCT (52.2% versus 62.9%) and the median LOT before daratumumab was higher (3 versus 1). Importantly, the follow-up in our cohort was considerably shorter (17.8 versus 44.3 months) and more than 35% of our patients were still on Da at data cut-off. Of these patients, 79.1% were on treatment with Da-IMiD. We find that the differences in timing, percentage of patients with high-risk CA, and shorter follow-up are the most import factors for poorer outcome in our cohort of patients.

Patients treated in the CASTOR study, where daratumumab was combined with bortezomib and dexamethasone, had a better outcome compared to our cohort of patients treated with Da-PI [14, 15]. The median follow-up time in our study and the CASTOR study were similar (20.3 versus 19.4 months). The ORR to Da-PI in our cohort was lower (60.6% versus 83.8%) and the TNT was shorter (median 5.3 months versus median PFS 16.1 months). Patients in our cohort were older (median 70 versus 64 years), had higher ISS stage III disease (29.5% versus 23.5%), more patients had high-risk CA (36.0% versus 22.7%) and were heavily pre-treated with a median number of prior LOT of 4 versus 2 in the CASTOR study. It is worth noticing that in our cohort of patients, 93% had prior exposure to bortezomib compared to 67.3% in the CASTOR study. The difference in outcomes between our patients and the

outcomes reported in the CASTOR study can be explained by timing of treatment, previous exposure to multiple drugs and the number of patients with high-risk CA.

Toxicities accounted for discontinuation in 8.8% of all patients. Of these, infections were the most frequent toxicity, followed by neuropathy and bone marrow suppression. Toxicities to Da-mono and Da-IMiD were similar to those observed in clinical trials. We report that 15.5% of patients treated with Da-PI stopped treatment due to toxicity compared to 7.4% in the CASTOR study [14]. In contrast with the CASTOR study, in which discontinuations due to peripheral neuropathy occurred in 0.4% of patients, we found that the leading cause of discontinuation of Da-PI was peripheral neuropathy (5.6%). This difference in toxicity may be explained by a higher exposure to bortezomib before Da in our cohort.

## Conclusion

We find that the real-world outcomes of multiple myeloma patients treated with daratumumab are worse than the results of clinical trials. Outcomes achieved with daratumumab were best when daratumumab was used in combination with IMIDs and in early lines of therapy. Patients with high-risk CA had worse outcomes, but patients with amp1q had similar outcomes to standard-risk patients. The poorer clinical performance of daratumumab-based therapies in our cohort compared with the results of phase 2 and 3 studies may be explained by later timing of daratumumab, higher percentage of patients with high-risk CA and shorter follow-up in the real-world setting.

## Supporting information

**S1 File.**
(PDF)

## Author Contributions

**Conceptualization:** Agoston Gyula Szabo, Annette Juul Vangsted.

**Data curation:** Agoston Gyula Szabo, Mette Bøegh Levring, Birgitte Preiss, Carsten Helleberg, Marie Fredslund Breinholt, Emil Hermansen, Lise Mette Rahbek Gjerdrum, Søren Thorgaard Bønløkke, Katrine Nielsen, Eigil Kjeldsen, Katrine Fladeland Iversen, Elena Manuela Teodorescu, Marveh Dokhi, Eva Kurt, Casper Strandholdt, Mette Klarskov Andersen, Annette Juul Vangsted.

**Formal analysis:** Tobias Wirenfeldt Klausen, Annette Juul Vangsted.

**Funding acquisition:** Annette Juul Vangsted.

**Investigation:** Annette Juul Vangsted.

**Project administration:** Annette Juul Vangsted.

**Writing – original draft:** Annette Juul Vangsted.

**Writing – review & editing:** Agoston Gyula Szabo, Tobias Wirenfeldt Klausen, Mette Bøegh Levring, Birgitte Preiss, Carsten Helleberg, Marie Fredslund Breinholt, Emil Hermansen, Lise Mette Rahbek Gjerdrum, Søren Thorgaard Bønløkke, Katrine Nielsen, Eigil Kjeldsen, Katrine Fladeland Iversen, Elena Manuela Teodorescu, Marveh Dokhi, Eva Kurt, Casper Strandholdt, Mette Klarskov Andersen, Annette Juul Vangsted.

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
