## [Decision Letter · Decision Letter 0]

27 Apr 2021

PONE-D-21-09072

The real-world outcomes of multiple myeloma patients treated with daratumumab

PLOS ONE

Dear Dr. Vangsted,

Thank you for submitting your manuscript to PLOS ONE. After careful consideration, we feel that it has merit but does not fully meet PLOS ONE’s publication criteria as it currently stands. Therefore, we invite you to submit a revised version of the manuscript that addresses the points raised during the review process, and in particular those from  by Reviewer #2.

We look forward to receiving your revised manuscript.

Kind regards,

Francesco Bertolini, MD, PhD

Academic Editor

PLOS ONE

Journal Requirements:

In your ethics statement in the Methods section and in the online submission form, please provide additional information about the data used in your retrospective study. Specifically, please ensure that you have discussed whether all data were fully anonymized before you accessed them and/or whether the IRB or ethics committee waived the requirement for informed consent. If patients provided informed written consent to have data from their medical records used in research, please include this information.

3. In your Methods section, please provide additional information about the participant selection methods. Please ensure you have provided sufficient details to replicate the analyses such as:

a) the full names of the participating institutions/departments,

b) a description of any inclusion/exclusion criteria that were applied to participant selection,

c) a statement as to whether your sample can be considered representative of a larger population

The work was funded by the Danish Cancer Society R249-A14646-19-S70, and Holms Mindelegat

„No”

Additionally, because some of your funding information pertains to commercial funding, we ask you to provide an updated Competing Interests statement, declaring all sources of commercial funding.

In your Competing Interests statement, please confirm that your commercial funding does not alter your adherence to PLOS ONE Editorial policies and criteria by including the following statement: "This does not alter our adherence to PLOS ONE policies on sharing data and materials.” as detailed online in our guide for authors  http://journals.plos.org/plosone/s/competing-interests.  If this statement is not true and your adherence to PLOS policies on sharing data and materials is altered, please explain how.

Please include the updated Competing Interests Statement and Funding Statement in your cover letter. We will change the online submission form on your behalf.

„no”

Reviewers' comments:

Reviewer's Responses to Questions

**Comments to the Author**

1. Is the manuscript technically sound, and do the data support the conclusions?

Reviewer #1: Yes

Reviewer #2: Yes

2. Has the statistical analysis been performed appropriately and rigorously? 

Reviewer #1: Yes

Reviewer #2: Yes

3. Have the authors made all data underlying the findings in their manuscript fully available?

Reviewer #1: Yes

Reviewer #2: Yes

4. Is the manuscript presented in an intelligible fashion and written in standard English?

Reviewer #1: Yes

Reviewer #2: Yes

5. Review Comments to the Author

Reviewer #1: The Authors report real-world outcomes of all Danish patients (n= 635) with multiple myeloma (MM) treated with daratumumab-based regimens until 1 January 2019. Information was collected retrospectively and included lines of therapy (LOT), hematologic responses according to the International Myeloma Working Group recommendations, time to next treatment (TNT) and the cause of discontinuation of treatment.

The median number of LOT given before daratumumab was 5 for Da-mono, 3 for Da-IMiD, 4 for Da-PI, and 2 for Da-other. In Da-mono, overall response rate (ORR) was 44.9% and median time to next treatment (mTNT) was 4.9 months. In Da-IMiD, ORR was 80.5%, and mTNT was 16.1 months. In Da-PI, OOR was 60.6% and mTNT was 5.3 months. In patients treated with Da-other, OOR was 54,2% and mTNT was 5.6 months. Early use of daratumumab was associated with longer TNT (p�0.0001).

The Authors conclude that the real-world outcomes of multiple myeloma patients treated with daratumumab are worse than the results of clinical trials. Outcomes achieved with daratumumab were best when daratumumab was used in combination with IMIDs and in early lines of therapy.

The study design is appropriate and includes a correct statistical analsysis. The manuscript is clearly written.

Reviewer #2: Szabo et al described a national experience with daratumumab alone or in combination with other drugs for the treatment of patients with multiple myeloma. They included a series of 635 patients, with 75% of them with cytogenetic data available. In a global perspective, this study draws attention to the real world results of one of the most used drug for multiple myeloma in Western countries. They also describe the importance of bias selection when including a patient in a clinical trial vs. those in the real clinical practice. With the limitations of registry study, they have built a clear manuscript for hematology community in other countries where daratumumab is approved and commercialized.

Some issues could improve the article. Some major and minor points over the manuscript are:

1. Overall survival results should be showed and included. In relapsed/refractory disease, quality of life and survival are the two main objectives. Showing only time to next treatment is a big limitation of the study and the real impact of daratumumab could be biased. A impact beyond PFS and TNT in terms of OS has been found in the original trials and the OS could help to

explore is that is also happening in real world practice.

2. The authors are dealing mainly with refractory and relapsed multiple myeloma. That should be stated in the title, the abstract, the methods section and elsewhere. Patients in first line should not be included in this manuscript as it seems not approved in Denmark at that time.

3. "Timing to treatment" in abstract and results section should be defined. It is related to the number of previous lines of treatment, that would be easier. The other option is the time from diagnosis. But it should be stated more clearly.

4. Are patients from clinical trials excluded in this study? This should be clarified in the methods section.

5. Novelty statement should be rewrite, including the number of patients studied and the number of patients with cytogenetic data available.

6. amp1q results are intriguing. As available with the same technique, the authors should try to include the information about del1p. Also, the power of amp1q plus other abnormalities vs. the amp1q by itself could be also of interest.

7. A chart of the approved treatment during that period of time in Denmark for multiple myeloma in first line and at relapse should be included.

8. "Lenalidomide maintenance" (page 6 line 111): it should be described "after ASCT".

9. The authors considered at least 10% of cytogenetic abnormalities for prognostication. However, the level is quite in debate in the myeloma community, including even 55% of del17p in some big series. Exploration of other cutoff could be useful, particularly with amp1q or even del17p.

10. Page 10, line 185: Da-mono is nor a "combination".

11. What regimens were available after 4th LOT for patients with MM vs. daratumumab in monotherapy? (otherwise, 2., 3., 4. system is confused)

12. Peripheral neuropathy (16%!) as a toxicity related to daratumumab is quite inconsistent. Authors should describe how many patients developed this complications outside bortezomib treatment. I know that in a registry study is difficult to analyze what was the drug associated with the problem; however, if almost all the cases are in Dara-bortezomib group, it should be stated. Was Bortezomib administered s.c. or i.v.?

13. Table 1 and Table 2 (and even Table 3)are extremely busy. They should be moved to supplementary data and only a summary should b showed in the manuscript. Figures 2B, 2C and 2D could be also moved to supplementary data.

14. Figure 3 should be clearer. Combining timing and CA abnormalities should be more to supplementary data. Here, a clear data with the impact of CA according to 2 or 3 groups could be more useful graphically.

15..

6. PLOS authors have the option to publish the peer review history of their article (what does this mean?). If published, this will include your full peer review and any attached files.

Reviewer #1: **Yes: **Carmelo Carlo-Stella, MD

Reviewer #2: No

---

## [Author Response · Author response to Decision Letter 0]

1 Jun 2021

Dear Francesco Bertolini, MD, PhD

Academic Editor

PLOS ONE

Thank you the constructive reviewer´s comments, and the opportunity to revise our manuscript.

I here send you the revised manuscript. Included is one copy where all changes are highlighted and one clean copy, and below I have commented on the reviewer´s criticism. The changes in response to the reviewers are highlighted in colour. 

We thank the reviewers for the suggestions that have improved the manuscript considerably.

1. Overall survival results should be showed and included. In relapsed/refractory disease, quality of life and survival are the two main objectives. Showing only time to next treatment is a big limitation of the study and the real impact of daratumumab could be biased. A impact beyond PFS and TNT in terms of OS has been found in the original trials and the OS could help to

explore is that is also happening in real world practice. We agree with the reviewer that OS is important. We have inserted OS in Figure 1 and in the text. However, even in real life it requires a long follow up, which will result in data that are uninteresting as we now move into the field of treatment with BiTes and Celmods. Our aim with this publication is to present TNT for daratumumab. Figure 1A show the TNT for the same groups and has not been changed.

We have included OS for

1. Dara + IMids+ dex

2. Dara + Btz + dex

3. Dara + dex

4. Other

from date of start of daratumumab treatment. The data are presented in the text and in figure 1 B. We have calculated the significance in OS between the groups. We have calculated median OS for all patients and inserted in the Table 1 and in in the text. 

2. The authors are dealing mainly with refractory and relapsed multiple myeloma. That should be stated in the title, the abstract, the methods section and elsewhere. Patients in first line should not be included in this manuscript as it seems not approved in Denmark at that time. Our aim was to include all patient treated with daratumumab nationwide within a given time period and therefore 10 patients (1.6%) treated with daratumumab at first line within clinical trials are included. We believe that it is fair to include these patients and include them in one figure to illustrate that the TNT is longer for patients treated in first line. Inclusion of the 10 patients treated with daratumumab as first line dose not change our conclusions.

3. "Timing to treatment" in abstract and results section should be defined. It is related to the number of previous lines of treatment, that would be easier. The other option is the time from diagnosis. But it should be stated more clearly. Timing of treatment is defined as timing of daratumumab in the sequence of all LOT that the patients received throughout the course of their disease. The sentence is inserted in the abstract as well as in the method section.

4. Are patients from clinical trials excluded in this study? This should be clarified in the methods section. 10 patients (1.6%) treated with daratumumab at first line within clinical trials are included. We kindly refer to question no 2.

5. Novelty statement should be rewrite, including the number of patients studied and the number of patients with cytogenetic data available. We have included the number of patients in the study and the number of patients with available cytogenetic analysis.

6. amp1q results are intriguing. As available with the same technique, the authors should try to include the information about del1p. Also, the power of amp1q plus other abnormalities vs. the amp1q by itself could be also of interest. This is a very interesting question. Unfortunately, analysis of del1p is not routine practice in Denmark and we cannot provide the results. However, we have included the power of amp1q plus other abnormalities vs. the amp1q. The results are inserted in the text and in supplementary Table 6. The following text is inserted at line 227: but TNT of patient with am1q was longer as compared to patients carrying high-risk CA combined and amp1q (P=0.036; Supplementary Table 9)

Amp1q 75 9.8 6.5-16.9 1 

High-risk + amp1q 55 5.6 3.6-12.4 1.56 (1.03-2.35) 0.036

7. A chart of the approved treatment during that period of time in Denmark for multiple myeloma in first line and at relapse should be included. A chart has been included as supplementary Table X

8. "Lenalidomide maintenance" (page 6 line 111): it should be described "after ASCT". Thanks for the precision, we have inserted after ASCT in the text as suggested.

9. The authors considered at least 10% of cytogenetic abnormalities for prognostication. However, the level is quite in debate in the myeloma community, including even 55% of del17p in some big series. Exploration of other cutoff could be useful, particularly with amp1q or even del17p. We agree with the referee. It is debated in the myeloma community, and many earlier trials do not even include the cut-off of the analysis. Some authors claim that it is irrelevant whether the cut-off is 10 or 60%. The argument is that if the treatment does not work in patients with f. ex. del17p this clone will be the prevailing clone at relapse. 

10. Page 10, line 185: Da-mono is nor a "combination". Thanks, we have changed the word “combination” to the word “regimen”

11. What regimens were available after 4th LOT for patients with MM vs. daratumumab in monotherapy? (otherwise, 2., 3., 4. system is confused) The available standard regimens used in the study period are shown in supplementary table 1 and Supplementary table XX, as reported above. The treating physicians could use the standard regimens or off-label combinations on a case-by-case basis.

We are not sure that we understand the question. Other treatment strategies include doctor´s choice of treatment 

12. Peripheral neuropathy (16%!) as a toxicity related to daratumumab is quite inconsistent. Authors should describe how many patients developed this complications outside bortezomib treatment. I know that in a registry study is difficult to analyze what was the drug associated with the problem; however, if almost all the cases are in Dara-bortezomib group, it should be stated. Was Bortezomib administered s.c. or i.v.? In this retrospective analysis toxicities were only registered if they were the cause of discontinuation of a line of therapy. This may have been unclear for the readers, and we therefore added an extra sentence to clarify this in methods and in the legends of Supplementary Table 15.

Supplementary Table 15 is to be viewed in relation to Supplementary Table 14. As shown in Supplementary Table 14, toxicity (in general) was the cause of discontinuation of the first daratumumab-containing line of therapy in 56 cases, which was 13.7% of all discontinuations. Supplementary Table 15 shows the type of toxicity in the 56 cases. Neuropathy was reported as the toxicity leading to discontinuation in 9 cases in total, across all daratumumab regimens, representing 16.1% of discontinuations due to toxicity. In total, this patient number represented 1.4% of the entire cohort of daratumumab-treated patients. The distribution of the 9 cases of neuropathy across the daratumumab regimens is shown in this table: 1 patient in Da-mono, 4 patients in Da-IMiD, 4 patients in Da-PI.

Although this was not registered specifically, all patient in this study were treated with subcutaneous bortezomib, which has been the standard of care in Denmark since 2011.

13. Table 1 and Table 2 (and even Table 3)are extremely busy. They should be moved to supplementary data and only a summary should b showed in the manuscript. Figures 2B, 2C and 2D could be also moved to supplementary data. We have prepared a summary of the results and moved the full tables as well as Figures 2B, 2C and 2D the supplementary material

14. Figure 3 should be clearer. Combining timing and CA abnormalities should be more to supplementary data. Here, a clear data with the impact of CA according to 2 or 3 groups could be more useful graphically. 

We understand that the figure may be busy, but we find that it is important to show standard risk as well as TNT for patients with high risk cytogenetic +/- amp1q and amp1q alone. We have removed patients with missing. 

TNT standard 

TNT for patient with high risk cytogenetic t(4;14); t(14;16); del(17p) and amp1q

TNT for patients amp1q

TNT for high risk cytogenetic t(4;14); t(14;16); del(17p)

We have removed the figures combining timing and CA abnormalities to supplementary data

---

## [Decision Letter · Decision Letter 1]

30 Jun 2021

PONE-D-21-09072R1

The real-world outcomes of multiple myeloma patients treated with daratumumab

PLOS ONE

Dear Dr. Vangsted,

Thank you for submitting your manuscript to PLOS ONE. After careful consideration, we feel that it has merit but does not fully meet PLOS ONE’s publication criteria as it currently stands. Therefore, we invite you to submit a revised version of the manuscript that addresses the points raised during the review process by Reviewer #2.

We look forward to receiving your revised manuscript.

Kind regards,

Francesco Bertolini, MD, PhD

Academic Editor

PLOS ONE

Journal Requirements:

Additional Editor Comments (if provided):

Reviewers' comments:

Reviewer's Responses to Questions

**Comments to the Author**

1. If the authors have adequately addressed your comments raised in a previous round of review and you feel that this manuscript is now acceptable for publication, you may indicate that here to bypass the “Comments to the Author” section, enter your conflict of interest statement in the “Confidential to Editor” section, and submit your "Accept" recommendation.

Reviewer #1: All comments have been addressed

Reviewer #2: (No Response)

2. Is the manuscript technically sound, and do the data support the conclusions?

Reviewer #1: (No Response)

Reviewer #2: Yes

3. Has the statistical analysis been performed appropriately and rigorously? 

Reviewer #1: (No Response)

Reviewer #2: Yes

4. Have the authors made all data underlying the findings in their manuscript fully available?

Reviewer #1: (No Response)

Reviewer #2: Yes

5. Is the manuscript presented in an intelligible fashion and written in standard English?

Reviewer #1: (No Response)

Reviewer #2: Yes

6. Review Comments to the Author

Reviewer #1: (No Response)

Reviewer #2: The authors have included many of the suggestions and modified the vast majority of critical points arisen during the first revision of the manuscript. Particularly, the inclusion of overall survival analysis and some methodological modifications. It seems much clearer now for me and I think for the readers too.

I am still worried about the 10 patients included in first line and within a clinical trial. I think even you could mention these patients in the manuscript, they are exceptions of the inclusion criteria in your study and, instead of improving the quality, this information diverts from the main goal of the paper.

7. PLOS authors have the option to publish the peer review history of their article (what does this mean?). If published, this will include your full peer review and any attached files.

Reviewer #1: No

Reviewer #2: No

---

## [Author Response · Author response to Decision Letter 1]

13 Sep 2021

Dear Francesco Bertolini, MD, PhD

Academic Editor

PLOS ONE

Thank you the reviewer´s comments, and the opportunity to revise our manuscript.

I here send you the revised manuscript. Included is one copy where all changes are highlighted and one clean copy, and below I have commented on the reviewer´s criticism. The changes in response to the reviewers are highlighted in colour. 

Reviewer #2: The authors have included many of the suggestions and modified the vast majority of critical points arisen during the first revision of the manuscript. Particularly, the inclusion of overall survival analysis and some methodological modifications. It seems much clearer now for me and I think for the readers too.

I am still worried about the 10 patients included in first line and within a clinical trial. I think even you could mention these patients in the manuscript, they are exceptions of the inclusion criteria in your study and, instead of improving the quality, this information diverts from the main goal of the paper.

We understand the concern from the reviewer´s and we thank you for your comments. Our inclusion criteria were treatment of daratumumab irrespectively of line of therapy. We have inserted a comment about these patients in the results. 

Ten patients received daratumumab in combination with lenalidomide in first line as part of the Maia study. The outcome of these patients is presented in figure 2.

---

## [Decision Letter · Decision Letter 2]

29 Sep 2021

The real-world outcomes of multiple myeloma patients treated with daratumumab

PONE-D-21-09072R2

Dear Dr. Vangsted,

We’re pleased to inform you that your manuscript has been judged scientifically suitable for publication and will be formally accepted for publication once it meets all outstanding technical requirements.

Kind regards,

Francesco Bertolini, MD, PhD

Academic Editor

PLOS ONE

Additional Editor Comments (optional):

Reviewers' comments:

Reviewer's Responses to Questions

**Comments to the Author**

1. If the authors have adequately addressed your comments raised in a previous round of review and you feel that this manuscript is now acceptable for publication, you may indicate that here to bypass the “Comments to the Author” section, enter your conflict of interest statement in the “Confidential to Editor” section, and submit your "Accept" recommendation.

Reviewer #2: (No Response)

2. Is the manuscript technically sound, and do the data support the conclusions?

Reviewer #2: Yes

3. Has the statistical analysis been performed appropriately and rigorously? 

Reviewer #2: Yes

4. Have the authors made all data underlying the findings in their manuscript fully available?

Reviewer #2: Yes

5. Is the manuscript presented in an intelligible fashion and written in standard English?

Reviewer #2: Yes

6. Review Comments to the Author

Reviewer #2: The authors reported "Ten patients received daratumumab in combination with lenalidomide in first line as part of the Maia study. The outcome of these patients is presented in figure 2." in the new version of the manuscript. However, as stated in the abstract and in the introduction "we report real-world outcomes of all Danish patients with multiple myeloma (MM) treated with daratumumab-based regimens until January 2019.". Results coming from MAIA trials are not exactly from "real-world data". In fact, in the conclusion they stated again: "The real-world outcomes of multiple myeloma patients treated with daratumumab are worse than the results of clinical trials." I do not think that it is coherent to add patients in first line treated in a clinical trial in a series of real-world data, mainly relapsed/refractory.

7. PLOS authors have the option to publish the peer review history of their article (what does this mean?). If published, this will include your full peer review and any attached files.

Reviewer #2: No

---

## [Editor Report · Acceptance letter]

4 Oct 2021

PONE-D-21-09072R2 

The real-world outcomes of multiple myeloma patients treated with daratumumab 

Dear Dr. Vangsted:

I'm pleased to inform you that your manuscript has been deemed suitable for publication in PLOS ONE. Congratulations! Your manuscript is now with our production department. 

Kind regards, 

on behalf of

Dr. Francesco Bertolini 

Academic Editor

PLOS ONE